# pH-Responsive Hydrogel Beads Based on Alginate, κ-Carrageenan and Poloxamer for Enhanced Curcumin, Natural Bioactive Compound, Encapsulation and Controlled Release Efficiency

**DOI:** 10.3390/molecules27134045

**Published:** 2022-06-23

**Authors:** Katarina S. Postolović, Milan D. Antonijević, Biljana Ljujić, Marina Miletić Kovačević, Marina Gazdić Janković, Zorka D. Stanić

**Affiliations:** 1Faculty of Science, University of Kragujevac, 34000 Kragujevac, Serbia; katarina.postolovic@pmf.kg.ac.rs; 2Faculty of Engineering and Science, School of Science, Medway Campus, University of Greenwich, Chatham Maritime, Kent ME4 4TB, UK; m.antonijevic@greenwich.ac.uk; 3Department of Genetics, Faculty of Medical Sciences, University of Kragujevac, 34000 Kragujevac, Serbia; bljujic74@gmail.com (B.L.); marinagazdic87@gmail.com (M.G.J.); 4Department of Histology and Embryology, Faculty of Medical Sciences, University of Kragujevac, 34000 Kragujevac, Serbia; marina84kv@gmail.com

**Keywords:** curcumin, hydrogel beads, alginate, κ-carrageenan, poloxamer, controlled release, oral delivery

## Abstract

Polyphenolic compounds are used for treating various diseases due to their antioxidant and anticancer properties. However, utilization of hydrophobic compounds is limited due to their low bioavailability. In order to achieve a greater application of hydrophobic bioactive compounds, hydrogel beads based on biopolymers can be used as carriers for their enhanced incorporation and controlled delivery. In this study, beads based on the biopolymers-κ-carrageenan, sodium alginate and poloxamer 407 were prepared for encapsulation of curcumin. The prepared beads were characterized using IR, SEM, TGA and DSC. The curcumin encapsulation efficiency in the developed beads was 95.74 ± 2.24%. The release kinetics of the curcumin was monitored in systems that simulate the oral delivery (pH 1.2 and 7.4) of curcumin. The drug release profiles of the prepared beads with curcumin indicated that the curcumin release was significantly increased compared with the dissolution of curcumin itself. The cumulative release of curcumin from the beads was achieved within 24 h, with a final release rate of 12.07% (gastric fluid) as well as 81.93% (intestinal fluid). Both the in vitro and in vivo studies showed that new hydrogel beads based on carbohydrates and poloxamer improved curcumin’s bioavailability, and they can be used as powerful carriers for the oral delivery of different hydrophobic nutraceuticals.

## 1. Introduction

Curcumin (Cur), also known as turmeric, is isolated from the roots of the *Curcuma longa* plant and belongs to the group of hydrophobic polyphenolic compounds. It possesses excellent antioxidant, anti-inflammatory, antibacterial and anticancer properties, and it is the subject of many research studies [1,2,3,4]. Despite the high efficacy of curcumin, its use for the prevention and treatment of many types of diseases (malignancies, arthritis, Alzheimer’s disease, diabetes, allergies and other chronic diseases) is limited due to its very low bioavailability [1,2,3,4]. Additionally, orally administered curcumin was found to undergo first-pass metabolism [2,5]. Within the gastrointestinal tract, various enzymatic reactions including curcumin lead to its transformation and the formation of its derivatives. In this way, the largest amount of ingested curcumin does not reach the bloodstream in its original form and is excreted through bile or feces [2]. Therefore, a more frequent daily intake of the drug is necessary. In order to increase its stability and oral bioavailability, different formulation strategies were developed and studied. Most frequently, nanoparticles [6], liposomes [7], micelles [8], phospholipid complexes [9], conjugates [10], solid lipid nanoparticles [11], polysaccharide-based complex particles [12,13] and hydrogel beads [14,15] were used as curcumin carriers.

Hydrogels represent three-dimensional, cross-linked polymers that are water-soluble and can be prepared in the form of beads, nanoparticles and films [16]. Due to the properties of the components which constitute them, as well as the way they are attached, hydrogels can be widely used as drug carriers [16]. The porosity of the hydrogels allows incorporation of the drug within the matrix and its subsequent controlled release [16]. Beads, nanoparticles and nanoliposomes, based on biopolymers, are widely used as systems for the transfer and delivery of drugs and nutraceuticals, thereby enhancing their utilization and reducing side effects [17,18,19]. The advantage of these carriers is reflected in their small size, slow degradation and possibility of controlled release of the drug [17,18,19]. Two natural biopolymers that are widely used in the pharmaceutical and food industries are alginate and carrageenan.

Carrageenan (Car) is a polysaccharide that is made up of galactose and anhydrogalactose, linked by a glycosidic bond and derived from red algae [20]. Depending on the way the sugar units are connected, as well as the number and positions of the sulfate groups, different types of carrageenan are distinguished, of which κ-, ι- and λ-carrageenan are of paramount importance in pharmacy [20]. The ability to build hydrogels involves κ- and ι-carrageenan, and in the presence of certain cations, especially potassium ions, its ability to form a gel increases, especially in the case of κ-carrageenan [20]. Although carrageenan itself has anticancer and anticoagulant properties, it is most commonly used as a carrier for various drugs. Additionally, much better properties demonstrate carriers in which carrageenan is cross-linked with other polymers [21,22].

Alginates (Alg, salts of alginic acid) are obtained from seaweed and represent polymeric polysaccharide, composed of D-mannuronic and L-guluronic acid chains [23]. Alginate also has the ability to build a hydrogel, which is used for encapsulation of different types of compounds [23]. In the presence of divalent and polyvalent metal ions, especially in the presence of calcium ions, a hydrogel can be formed. This crosslinking process of alginates, usually formed by the interaction of guluronic acid chains and calcium ions, is best described using the “egg-box” model [24]. However, the presence of divalent and polyvalent metal ions, depending on their concentrations, can lead to the formation of insoluble alginate hydrogel precipitate [23,25]. For this reason, the adequate selection of a cation and its concentration is vital to the proper formation of the desired product [23,24]. Calcium and sodium alginates are non-toxic and biocompatible, being used in the pharmaceutical industry as carriers for cell immobilization or encapsulation of bioactive molecules and drugs [23,24]. These incorporated substances can be released at the desired rate and kinetics in a controlled release system.

The most common way of administering medications is orally. However, in some cases, there is a rapid passage of the drug through the body, and usually, undesirable enzymatic degradation of the drug can occur in the gastrointestinal tract as well. Therefore, great attention is given to pH-responsive hydrogels that can be used in the pharmaceutical industry as carriers for the controlled release of drugs in a medium of a desired pH value [26,27,28]. pH-responsive hydrogels are high-molecular polymers that undergo a volume or phase transition when the pH value of the external environment changes. In addition, the polymers comprising these hydrogels contain groups such as carboxyl, amino and sulphate, which can be protonated or deprotonated depending on the pH value [26,27,28]. Thanks to these properties, appropriate polymer selection can affect the swelling of hydrogels in mediums of different pH values and thus influence the drug release profile in the desired mediums [26,27,28]. The cross-linking of carrageenan and alginate can form a bead-shaped hydrogel that can be used as a carrier for incorporation and later the controlled release of different compounds [29,30,31].

Various carriers based on polysaccharides (e.g., carrageenan, alginate, chitosan and cellulose) and modifications thereof have been tested, and previous studies have reported that the bioavailability of curcumin and its controlled release can be improved by the encapsulation of curcumin in these carriers [6,7,10,12,13,14,32,33,34]. In this study, beads based on cross-linked carrageenan and alginate were developed, and curcumin was used as a drug model. Since curcumin is a highly hydrophobic drug, poloxamer 407 was also included in the bead composition in order to improve curcumin solubility. Poloxamer 407 is a non-ionic surfactant and represents a copolymerized ethylene and propylene oxide that also has the property of forming hydrogels [35,36]. Its benefits are reflected in the increase in drug solubility, metabolic stability and nontoxicity and the ability to deliver the drug more selectively to cancerous cells [35,37,38,39]. Considering the importance of curcumin and the necessity of increasing its bioavailability, the characteristics of beads based on the cross-linked polymers carrageenan, alginate and poloxamer for the controlled release of curcumin were evaluated. The effect of newly synthesized beads that contain incorporated curcumin was tested in vitro by monitoring the release of curcumin under conditions which simulated its intake by the oral route. Finally, the in vivo oral administration of curcumin was evaluated in mice for a curcumin aqueous suspension and curcumin-incorporated beads.

## 2. Results and Discussion

### 2.1. Optimization of Beads’ Content

The basic criteria which have been considered during the optimization process of the bead composition are the speed of formation, surface area and shape of the beads, swelling degree of the non-drug beads and curcumin encapsulation efficiency.

#### 2.1.1. Carrageenan/Alginate Beads

It is known that in the presence of Ca^2+^ ions, cross-linking of alginates occurs due to the interchain interaction of polysaccharides, which results in the formation of beads of alginate-Ca^2+^. Similarly, the presence of K^+^ ions leads to cross-linking of carrageenan, wherein the K^+^ ion interacts with a sulphate ester and oxygen atoms on the polysaccharide. After cross-linking of alginate-Ca^2+^ and carrageenan-K^+^, it is considered that this further leads to their interweaving, thus forming a unique, cross-linked hydrogel polymer [31].

To obtain the optimum composition of the beads for the formation of spherical shape particles with an approximate size, through the interaction of the corresponding saccharides in different ratios in the presence of Ca^2+^ and K^+^ ions, carrageenan/alginate beads were made (Figure 1a). The ratios 6:4 and 8:2 of κ-Car/Alg were excluded from further experimentation, as high concentrations of carrageenan created a very thick mass unsuitable for adding in a drop form, which further prevented the formation of beads.

The degree of swelling is a very important parameter, indicating the effectiveness of controlled drug release from the carrier, namely the beads. The swelling of the hydrogel indicates on series of interactions that result in increasing distances between the cross-linked polymer chains. Increasing the distance allows the release of a drug that is incorporated [40]. Figure 2 shows the time dependence of the swelling degree for beads containing different carrageenan/alginate ratios at different pH values (pH 1.2 and pH 7.4).

The swelling degree of the beads was significantly higher in the pH 7.4 buffer than in an acidic medium. This could be explained by the presence of a great number of carboxylate and sulphate anions originating from the alginate and carrageenan, respectively. In pH 1.2 hydrochloric acid, these anions are mainly protonated and exist in the –COOH and –OSO_3_H forms. Hydrogen bonds can easily be formed between the protonated forms of carrageenan and alginate. These interactions between polymers can be stronger than the interactions polymers will achieve with water from the solution during the swelling process [41]. However, in a PBS buffer (pH 7.4), the dominant form of carrageenan and alginate will be anionic, and their mutual rejection will cause an increase in their distance. In addition, hydrogen bonds between the water and anionic forms of saccharides can easily be formed, leading to water absorption by the hydrogels (i.e., swelling of the beads).

Based on the obtained results, it can be seen that higher alginate concentrations led to a swelling degree decrease in the acidic mediums. Similar final swelling degree values were obtained in the PBS buffer for Car/Alg beads with different saccharide ratios. As the primary purpose of using the prepared beads is the oral transport of curcumin (which degrades quickly in an acidic medium), the beads with the lowest swelling rate in an acidic medium were used in the following experiment. Additionally, during the bead preparation process, it was noticed that the higher concentration of alginates led to easier bead formation and greater weight for the obtained dry beads. Based on these criteria, it was concluded that the optimal carrageenan/alginate ratio was 2:8, and this saccharide ratio was used in further experimental work.

#### 2.1.2. Carrageenan/Alginate-Curcumin Beads

Prior to the introduction of curcumin into polysaccharides in the presence of surfactant, the properties of curcumin in a mixture of polysaccharides were evaluated (Figure 1b). During the production of beads, the formation of spherical beads did not occur in the early stage of adding drops but rather the formation of emulsified droplets that made aggregates on the surface of the liquid, most probably as a result of the addition of a larger volume of ethanol (5.0 mL) in the process of dissolution of the curcumin (see experiment procedure). It was established [42] that hydrogel beads are formed, in most cases, by trapping a certain amount of water via different interactions (molecular entanglements, hydrogen bonding, hydrophobic forces, etc.). Luo et al. [42] concluded that solvent composition plays an important role during hydrogel bead formation. If the volume of the added ethanol increases, hydrogen bonding between water and ethanol will increase, and consequently, the hydrogen bonds between saccharides and water, necessary for the formation of spherically shaped beads, will be broken. As a consequence, the yield of the obtained dried beads was lower compared with the beads prepared containing poloxamer.

#### 2.1.3. Carrageenan/Alginate/Poloxamer and Carrageenan/Alginate/Poloxamer-Curcumin Beads

Thus far, the incorporation and release of the hydrophilic drug betamethasone acetate from the hydrophilic beads based on carrageenan and alginate has been investigated [30]. As curcumin (the main objective of this study) is a highly hydrophobic drug, beads including poloxamer 407 along with saccharides were prepared. The addition of surfactant, a poloxamer, was aimed at improving the solubility of curcumin and, therefore, its bioavailability, as was previously demonstrated in [43].

In our work, beads containing poloxamer (Figure 1c) as well as poloxamer and curcumin (Figure 1d) were successfully prepared. It can be noticed that the poloxamer-containing beads were different compared with the poloxamer-free beads (Figure 1a,c vs. Figure 1b,d).

Poloxamer concentrations below 2.5% were insufficient to dissolve the total quantity of curcumin during the preparation of poloxamer/curcumin-containing beads. Therefore, adding a larger volume of ethanol was necessary, which led to the formation of undesirable emulsified drops at the beginning of instillation. Consequently, the obtained beads’ yield was slightly lower compared with that of the beads containing higher poloxamer concentrations. For this reason, only beads with a poloxamer concentration of 2.5–15.0% were considered in the further experimental work.

The swelling degree of the beads containing different poloxamer concentrations was examined in pH 1.2 hydrochloric medium and pH 7.4 phosphate buffer (Figure 3).

As shown in Figure 3, the swelling degree of the Car/Alg/Pol beads, similar to the Car/Alg beads, in an acidic environment was significantly lower than the swelling degree of the beads in the buffer. It can also be noticed that increasing the concentration of the poloxamer increased the beads’ swelling degree at pH 1.2 but also led to a solubility increase in the beads in the PBS buffer. For this reason, monitoring the swelling process of beads with poloxamer concentrations above 2.5% was discontinued after 100 min. By comparing the swelling degree of the Car/Alg and Car/Alg/Pol beads, it can be seen that adding poloxamer reduced the swelling degree of the beads. This can be explained by the hydrogen bonds formed between the poloxamer and the carboxylate anion of alginate or the sulphate anion of carrageenan. As a result of forming hydrogen bonds, interactions with water that lead to swelling are difficult. Similar results were obtained in other studies where poloxamer was used as a constituent of the prepared formulation [44,45]. Due to the high solubility of the Car/Alg/Pol beads (15.0%) in the buffer, this concentration of poloxamer was excluded from further experimental work.

The percentage of encapsulation of curcumin in the Car/Alg- as well as Car/Alg/Pol-based beads with different poloxamer concentrations was determined. The percentage of encapsulation efficiency (EE (%)) in the beads is shown in Figure 4.

Although the swelling degree of the Car/Alg/Pol beads in the buffer was slightly lower than that of the Car/Alg beads, the advantage of beads containing poloxamer is that the encapsulation efficiency is much higher in the presence of a poloxamer compared with the beads not containing a surfactant. However, care must be taken to add an adequate poloxamer concentration. Increasing the poloxamer concentration (5.0% and 10.0%, *w/v*) leads to a decrease in the encapsulation efficiency. This may be the consequence of excessively added poloxamer because excess of poloxamer can pass from the beads to the CaCl_2_ and KCl solution for instillation (see the bead preparation procedure). In addition, a higher amount of poloxamer in the CaCl_2_ and KCl solution can increase curcumin diffusion from the beads into the solution during the bead preparation process. Therefore, the curcumin degree of encapsulation will be lower.

By comparing these results with the findings from other studies [10,43,46], the following advantages of the beads generated in this work (Car/Alg/Pol-Cur) can be observed. The percentage of curcumin encapsulation in the nanoparticles based on chitosan, alginate and poloxamer was between 10% and 13%, wherein the maximum added mass of curcumin was only 1 mg [43]. Dey and Sreenivasan [10] determined the mass of incorporated curcumin, and it was (1.09 ± 0.53) mg in 100 mg of the prepared conjugate, and in the current work, the mass of incorporated curcumin in 100 mg of the beads was 6.12 ± 0.14 mg. Finally, another group of authors [46] obtained a percentage of encapsulation to the amount of 78%, but this percentage was referring to the weight of the measured curcumin, which was equal to 2 mg. Therefore, it can be concluded in our experiment that the mass of the incorporated curcumin and its encapsulation efficiency were significantly higher (95.74% from 50 mg of Cur) compared with the results obtained by the above-mentioned authors.

Table 1 shows the average diameter of the obtained bead films, as well as the mass of incorporated curcumin in the beads of an optimal composition.

Based on all our analysis, the product formed by using saccharides at a ratio of 2:8, with the addition of 2.5% poloxamer 407 and 50 mg of curcumin, was deemed as the optimal formulation for testing the in vitro release of curcumin.

### 2.2. Characterization of Beads by FTIR

The basic FTIR spectra for the κ-carrageenan, alginate, poloxamer 407 and curcumin, as well as the Car/Alg, Car/Alg/Pol, Car/Alg-Cur and Car/Alg/Pol-Cur beads, are shown in Figure 5a,b. In the carrageenan and alginate spectra, wide absorption bands can be observed in the 3400–3200-cm*^−^*^1^ wavelength range, originating from the vibration of the O–H bond. In the alginate spectrum, two characteristic absorption bands can be noticed at 1590 and 1404 cm*^−^*^1^, and these were caused by asymmetric and symmetric vibrations of the carboxylate anion. The absorption band at 1230 cm*^−^*^1^ in the carrageenan spectrum was the result of vibrations of the sulphate groups. In the poloxamer spectrum, a band at 2875 cm*^−^*^1^ (the result of valence vibrations of C–H bonds) is observed, but there is also a very intense band at 1097 cm*^−^*^1^ (due to valence vibrations of the C–O bonds). In the curcumin spectrum, a sharp band at 3508 cm*^−^*^1^ was attributed to vibration of the phenolic O–H groups. Additionally, a band at 1628 cm*^−^*^1^ can be noticed, resulting from C=O stretching vibration.

The absorption bands in the FTIR spectrum of the carrageenan/alginate composite were at similar wavenumbers to those described in previous studies [31,40,47]. The FTIR spectrum contained all the vibrations of the groups that are characteristic of both carrageenan (sulphate group) and alginate (carboxylate anion), with a slight shift in the wavenumber values (1230→1236, 1404→1422 and 1590→1602 cm*^−^*^1^). It could be proposed that this change occurred due to the interaction between polysaccharides through intermolecular hydrogen bonding. In the Car/Alg/Pol spectrum, a very intensive and sharp band at 1080 cm*^−^*^1^, which was the result of the C–O bond vibration characteristic for poloxamer, confirms the presence of a poloxamer in the carrageenan/alginate mixture. In the Car/Alg-Cur spectrum, all the characteristic bands of the previously considered Car/Alg spectrum could be identified, with a minute band position change (1236→1248 and 1602→1596 cm*^−^*^1^) due to curcumin interaction with the carrageenan/alginate cross-linked polymer network beads. The presence of curcumin in the Car/Alg/Pol composite yielded a characteristic band shape and position changes. In the FTIR spectrum of the Car/Alg/Pol-Cur, after the loading of curcumin into blank Car/Alg/Pol, the band in the region of complex vibration of O–H groups (between 3200 cm*^−^*^1^ and 3400 cm*^−^*^1^) was transformed into a big, more intensive and sharper band at 3350 cm*^−^*^1^. Additionally, there was a slight shift in the bands corresponding to the vibration of alginate carboxylate anion (1422→1428 cm*^−^*^1^) and carrageenan sulphate anion (1236→1260 cm*^−^*^1^), which may have been the result of curcumin interaction with the carrier’s components.

### 2.3. SEM Analysis

The study of the shape, morphology and surface characteristics of the Car/Alg, Car/Alg/Pol, Car/Alg-Cur and Car/Alg/Pol-Cur beads was performed using scanning electron microscopy with different magnifications (Figure 6).

As can be seen from the presented results, adding poloxamer to the beads greatly altered their surface texture, as the surface of the Car/Alg and Car/Alg-Cur beads was rougher compared with that of the Car/Alg/Pol and Car/Alg/Pol-Cur beads. The Car/Alg beads formed an undulant and coarse structure with cylindrical and spherical solid-shaped aggregates (Figure 6a–a″). The presence of these aggregates can be attributed to excess K^+^ and Ca^2+^ ions added in the crosslinking process during the beads’ preparation but also to alginate and carrageenan blending or interaction between the alginate/carrageenan and K^+^/Ca^2+^ ions. The curcumin-loaded Car/Alg beads also exhibited rough surfaces, which was additionally attributed to curcumin’s presence on the surface (Figure 6b–b″). The shape of the aggregates on the surface of the Car/Alg-Cur beads was somewhat different from that on the Car/Alg beads, leading to the conclusion that a small part of curcumin was not incorporated inside the beads and remained on the surface. On the other hand, the Car/Alg/Pol and Car/Alg/Pol-Cur hydrogel beads showed a compact, smoother and tighter surface. A slightly more porous structure for the Car/Alg/Pol beads can be observed (Figure 6c–c″). The porous structure could generate the capillary forces that facilitated the penetration of fluids into the beads, resulting in bead swelling and drug release [48]. Finally, based on the SEM image of the remarkably smooth and compact Car/Alg/Pol-Cur beads surface (Figure 6d–d″), it can be concluded that curcumin was uniformly distributed inside the beads thanks to the formed interactions with both the saccharides and poloxamer.

### 2.4. Thermal Characteristics

#### 2.4.1. Thermogravimetric Study

TGA was used to investigate the thermal decomposition of the prepared samples. Figure 7a,b shows the thermograms of the applied compounds before and after forming different types of beads (Car/Alg, Car/Alg-Cur, Car/Alg/Pol and Car/Alg/Pol-Cur). The results showed that there were two significant weight loss stages for the biopolymeric compounds and composites in the TGA. The initial weight loss was caused by water loss (before 100 °C), indicating that the κ-carrageenan and alginate powder with a small size had a strong tendency to absorb water from the atmosphere. The second weight loss of the pure κ-carrageenan and alginate was around 210 °C (Alg) as well as 240 °C (Car) with a high mass loss rate, which was attributed to the thermal decomposition. Curcumin and poloxamer, as pure substances, do not have a tendency to absorb water from the atmosphere. The only mass loss was caused by degradation of the mentioned components and occurred at 260 °C and 310 °C for the curcumin and poloxamer, respectively.

Bead formation (Car/Alg and Car/Alg-Cur) led to decreased absorption of water from the atmosphere because the water loss prior to reaching 100 °C was negligible. Further, the weight loss that occurred at around 170 °C (Car/Alg and Car/Alg-Cur) as well as 190 °C (Car/Alg/Pol and Car/Alg/Pol-Cur) could be attributed to the release of water which was trapped during the beads’ formation (through interactions with the carboxylate and sulphate groups). Additionally, thermal decomposition of the formed beads occurred at a higher temperature (280 °C) compared with the thermal decomposition of pure κ-carrageenan and alginate, which indicates that the crosslinking of κ-carrageenan and alginate led to thermal stabilization of the hydrogel formulations. Furthermore, it can be concluded that the presence of a poloxamer in the Car/Alg/Pol beads led to the increase in the weakly bonded water content due to a higher mass loss (7.81%) under 100 °C in comparison with the Car/Alg beads (1.21%). On the other side, mass loss attributed to the evaporation of water, which was bonded during bead preparation, in the presence of a poloxamer occurred at a 20 °C higher temperature in comparison with the Car/Alg beads. Finally, based on the presented thermogravimetric curves, it can be concluded that the curcumin- and poloxamer-incorporated beads (Car/Alg/Pol-Cur) showed greater thermal stability compared with the beads containing only curcumin (Car/Alg-Cur) or a poloxamer (Car/Alg/Pol). Similar results were obtained by Rasool et al. [49] and Sun et al. [50].

#### 2.4.2. Differential Scanning Calorimetry

With the aim to study the phase transition property of the Cur-filled hydrogel beads, DSC was carried out at a temperature below 200 °C, as shown in Figure 8.

The DSC curve of the κ-carrageenan/alginate beads showed a single broad peak at about 90 °C, while the poloxamer and curcumin curves each showed a single sharp peak at 55 °C and 179 °C, respectively. The endothermic peak at 179 °C corresponds to the melting point of the crystalline curcumin [51]. However, there was no defined peak in the DSC traces of Car/Alg/Pol-Cur. This suggests that the curcumin had not kept its crystal structure in the beads; it was transformed into an amorphous state during the encapsulation process in the Car/Alg/Pol composite. Amorphous curcumin has a higher solubility and dissolution rate than its crystalline form. In addition, similar results have been reported in the literature for curcumin encapsulation in zein nanoparticles [52] and polyvinyl pyrrolidine nanoparticles [53], as well.

### 2.5. In Vitro Release of Curcumin

With the intention to determine whether there is an increase in the bioavailability of curcumin and its controlled release, the in vitro release of curcumin from the Car/Alg/Pol-Cur beads at different pH values was enacted. As a control, the release of pure curcumin itself was monitored under the same experimental conditions, whereby it was concluded that its concentration was negligible and not significantly changed over the course of time (figure not shown). The in vitro release of curcumin from the beads is shown in Figure 9.

Based on the obtained results for curcumin’s in vitro release under the simulated gastrointestinal tract conditions, it can be concluded that the pH value of the solution significantly conditioned the release of curcumin from the prepared carriers. The low curcumin release in pH 1.2 hydrochloric acid (12.07% after 24 h) could be caused by the low solubility of curcumin in acidic conditions and the swelling degree of the beads at pH 1.2 (Figure 3). Unlike the release at pH 1.2, the release of curcumin in the PBS buffer started with a modest, gradual increase (first 90 min) and then rose, reaching a significantly high jump after 6 h. Finally, the release percentage stabilized in the period from 7 to 24 h (81.93 ± 2.89% after 24 h). The significantly higher release percentage in the PBS buffer can be explained by the higher swelling degree of the beads at pH 7.4 (Figure 3) and the higher probability that the solution diffused within the beads, thus allowing curcumin release. In addition, hydrogel bead disintegration could be the cause of the high release percentage increase after 6 h. During the swelling process, phosphate ions from the PBS buffer penetrate the beads together with water molecules [15]. Over time, after achieving a high swelling degree, phosphate ions can react with cross-linked calcium alginate from the carrier and break the bonds which alginate carboxyl groups form with calcium ions because the affinity of calcium ions to phosphates is higher than that to alginate in a neutral medium. This leads to bead dissolution in the buffer, which further accelerates the release of curcumin in the PBS buffer. Similar drug release profiles from different formulations were obtained in other studies [15,54].

If the results obtained in our paper are compared with those of [55], where a polyvinyl alcohol-sodium alginate blend composited with 3D graphen oxide was analyzed, it can be concluded that better results were obtained in our paper. The percentage of curcumin release from the carrier for 35 h was about 25% [55], which is significantly less compared with the percentage of release achieved in our research (81.93%). Further comparison of the obtained results in our research with the results of another study [56] reveals the advantages of the prepared pH-responsive hydrogel beads. The curcumin release percentage from the polyethylene glycol-decorated graphene oxide nanosheets in the first 24 h was about 40%, and after 200 h, it reached a maximum value of 60% [56]. If the results of this study [56] are compared with our results, it can be concluded that in our study, better results were obtained in terms of the percentage and mass of released curcumin.

Based on the obtained results, it can be highlighted that a better curcumin release was achieved in the simulated intestinal fluid conditions, while a small percentage of curcumin was released in the simulated gastric fluid conditions. Additionally, the beads retained their shape in an acidic medium for 24 h. Curcumin-loaded beads administered by the oral route first reach the stomach, where a minimal amount of the drug is released. Then, the beads that have retained their shape and the highest percentage of incorporated curcumin can reach the intestinal fluid. Thus, curcumin’s rapid first-pass metabolism and its excretion in the bile and feces [3] are avoided, which is the most significant advantage of prepared hydrogel beads for oral drug administration.

### 2.6. The Kinetics of Release

To study the mechanism of curcumin release from the prepared beads, various mathematical equations were applied to the drug dissolution data (zero-order kinetics, first-order kinetics, the Higuchi model, Hixson–Crowell model, Baker–Lonsdale model, and Korsmeyer–Peppas model). The correlation coefficients (R^2^) and *k* and *n* values are shown in Table 2.

Based on regression coefficient analysis, the curcumin release from Car/Alg/Pol-Cur beads in an acidic medium and in a PBS buffer could be the best described using the Korsmeyer–Peppas model. The curcumin release from the Car/Alg/Pol-Cur beads in the PBS buffer could also be explained using the Hixon–Crowell model (R^2^ = 0.9358), which describes drug release mainly controlled by the diminishing surface of the drug particles during the dissolution process [57]. Additionally, the curcumin release in the PBS buffer, to a certain extent, could be fitted to the Highuchi and Baker–Lonsdale models, which are used to describe the drug release, controlled by Fick’s law of diffusion, as well as drug release from a spherical matrix, respectively [57].

In order to further study the drug release mechanism, the Korsmeyer–Peppas model was applied to determine the parameter that had the most significant impact on the release rate (polymer swelling, incorporated substance diffusion or polymer degradation) [57]. An obtained release exponent value of *n* = 0.43 for the spherical carriers indicates that the release is controlled by drug diffusion [57]. In contrast, a value of *n* = 0.93 indicates that drug release occurs primarily due to polymer swelling [57]. If the values of *n* differ from the above, then the release mechanism is influenced by several factors. Release exponent values below 0.43 correspond to a Fick’s law-controlled diffusion, and values above 0.43 correspond to anomalous release (i.e., release caused by diffusion and “erosion” of the polymers) [57]. Values above 0.93 correspond to super case II transport [57]. The values of *n* obtained during the release of curcumin in both the acidic medium (*n* = 1.002) and PBS buffer (*n* = 1.145) indicate that the release took place by super case II transport, where it was mainly conditioned by swelling and macromolecular relaxation of the polymeric chains.

### 2.7. Inflammation Study

Implantation of biomaterials is nearly always associated with inflammation, which is a biological response connected with activated inflammatory cells, such as macrophages. As a response to an infection, macrophages release toxic molecules such as nitrite. It is known that carrageenan in higher concentrations can induce inflammatory responses in laboratory animals [20]. Because of that, the inflammatory response to the prepared beads (Car/Alg/Pol and Car/Alg/Pol-Cur) was investigated using a nitric oxide assay (Figure 10).

In the current study, the results indicate that the Car/Alg/Pol and Car/Alg/Pol-Cur beads inhibited the LPS-induced nitrite production in RAW 264.7 macrophages during their treatment with hydrogel beads (10, 50 and 100 μg beads/mL). Due to the anti-inflammatory properties of curcumin [1,2,3], nitrite production in the presence of the Car/Alg/Pol-Cur beads was reduced more in comparison with the Car/Alg/Pol beads. Based on the obtained results, it can be concluded that the investigated materials are suitable for use as implantable carriers.

### 2.8. In Vivo Study

The in vivo bioavailability of the pure curcumin and prepared Car/Alg/Pol-Cur beads were determined by oral administration to rats at a dose of 50 mg/kg. The graph of dependence of the curcumin plasma concentration over time and the curcumin pharmacokinetic parameters are shown in Figure 11 and Table 3, respectively.

The C_max_ of the pure curcumin and Car/Alg/Pol-Cur beads was achieved after 2 h. Oral administration of the curcumin in the form of beads led to a higher and more prolonged level of curcumin in the blood of the mice (C_max_ = 219.7 ± 2.9 ng/mL; AUC_0–24 h_ = 2.91 μg·h/mL) compared with the crystalline form (C_max_ = 89.9 ± 2.9 ng/mL; AUC_0–24 h_ = 1.04 μg·h/mL). The use of Car/Alg/Pol-Cur beads enabled a stable curcumin plasma concentration from 0 to 24 h in comparison with the crystalline curcumin, for which the concentration decreased in the plasma over time. An increase in the curcumin concentration in the plasma of mice treated with Car/Alg/Pol-Cur beads in the period between 6 and 12 h could be related to the similar results obtained during the in vitro release study (Figure 9) due to disintegration of the beads (explained in Section 2.5.). In summary, the stable and increased curcumin plasma concentration indicates that the Car/Alg/Pol-Cur beads could enhance the pharmacological effects of curcumin.

## 3. Materials and Methods

### 3.1. Materials

Curcumin (Sigma Aldrich, Saint Louis, MO, USA), κ-carrageenan (Sigma Aldrich, Saint Louis, MO, USA, cat. No. 22048, molecular weight 672,000 g/mol and sulphate content 20.3%, according to [58]), sodium alginate (Sigma Aldrich, Saint Louis, MO, USA, cat. No. 180947, mannuronic acid to guluronic acid (M/G) ratio 1.56 and molecular weight 120,000–190,000 g/mol, according to the manufacturer), poloxamer 407 (Sigma Aldrich, Saint Louis, MO, USA), potassium chloride (Fisher Scientific, Waltham, MA, USA, laboratory reagent grade), calcium chloride dihydrate (Sigma Aldrich, Saint Louis, MO, USA, reagent plus ≥99.0%), anhydrous sodium hydrogen phosphate (Fisher Scientific, Waltham, MA, USA, laboratory reagent grade), potassium dihydrogen phosphate (Kemika, Zagreb, Croatia, laboratory reagent grade), sodium chloride (Sigma, Saint Louis, MO, USA, laboratory reagent grade), hydrochloric acid (~37%, Fisher Scientific, Waltham, MA, USA, laboratory reagent grade), fetal bovine serum (Sigma Aldrich, Saint Louis, MO, USA), DMEM (Sigma Aldrich, Saint Louis, MO, USA), LPS (Sigma Aldrich, Saint Louis, MO, USA), acetonitrile (Fisher Scientific, Waltham, MA, USA, HPLC grade), acetic acid (Fischer Scientific, Waltham, MA, USA, HPLC grade) and Griess reagent (Sigma Aldrich, Saint Louis, MO, USA) were used in this study.

### 3.2. Preparation of Carrageenan/Alginate Beads (Car/Alg)

The carrageenan/alginate beads were prepared using a method described previously [30,31]. Solutions of κ-carrageenan (3.5%, *w*/*w*) and sodium alginate (3.5%, *w*/*w*) were separately made by dissolving the corresponding amounts of saccharide in deionized water and heating the solution at 80 °C and 70 °C, respectively. The indicated temperatures were maintained for 30 min with constant stirring.

In order to determine the optimal composition of beads made of κ-carrageenan (κ-Car) and sodium alginate (Alg), beads containing different ratios of these saccharides (κ-Car/Alg (2:8, 4:6, 5:5, 6:4 and 8:2)) were prepared. The resulting mixtures, weighing 10 g, were stirred at 60 °C for 15 min. Subsequently, the mixtures were added dropwise to a solution of salt (200 mL 1.5%, *w*/*v* KCl and 1.5%, *w*/*v* CaCl_2_) at a temperature of 55 °C. In order to accelerate cross-linking, the solution, with the just-formed beads, was left for another 30 min at the same temperature and stirring conditions. Finally, the beads were washed and dried overnight in an oven at a temperature of 30 °C. The optimal ratio of carrageenan and alginate was determined and used in the further work.

### 3.3. Preparation of Carrageenan/Alginate/Poloxamer Beads (Car/Alg/Pol)

For the preparation of beads based on carrageenan, alginate and poloxamer, the following procedure was applied. The previously determined optimal amounts of alginate and κ-carrageenan were mixed, and the solution was heated at 60 °C for 15 min. Thereafter, 10.0 mL of aqueous poloxamer solution of different concentrations (1.0%, 2.5%, 5.0%, 10.0% and 15.0% *w*/*v*) was added, and stirring was continued for 30 min, maintaining the same temperature. The further preparation steps were the same as for the carrageenan/alginate beads.

### 3.4. Preparation of Curcumin-Incorporated Beads (Car/Alg-Cur and Car/Alg/Pol-Cur)

In order to prepare the carrageenan/alginate beads containing curcumin (Car/Alg-Cur), curcumin was dissolved in a mixture of ethanol and water (1:1), and a curcumin solution (5 mg/mL) was added to the alginate/κ-carrageenan mixture. The further course of experimental work was the same as for the preparation of carrageenan/alginate beads.

To prepare the beads containing poloxamer and curcumin (Car/Alg/Pol-Cur), curcumin solutions were prepared by dissolving curcumin in a poloxamer solution (for the 1% poloxamer solution, adding a small volume of ethanol was necessary). Then, the curcumin solution (5 mg/mL) was added to a mixture of carrageenan and alginate. The rest of the procedure was identical to the procedure for obtaining the carrageenan/alginate-containing beads.

### 3.5. Characterization of the Beads by FTIR and SEM

For determination of the composition of the beads and their characterization, Fourier-transform infrared (FTIR) spectra were recorded using an FTIR spectrophotometer (Perkin Elmer FTIR Spectrum Two, Waltham, MA, USA). The spectra were recorded in the wavenumber range of 4000–500 cm*^−^*^1^. In order to study the morphology of the beads and determine their shape and size, scanning electron microscopy (SEM, HITACHI SU8030, Tokyo, Japan, voltage source connected to a 1 kV accelerator, 10 µA current emitted) was used.

### 3.6. Thermal Characterization of Beads

The thermal characteristics of the prepared beads were determined using thermogravimetric analysis (TGA) and differential scanning calorimetry (DSC). Thermogravimetric analysis was performed using a TA instruments Q-5000 (Crawley, UK). The method involved heating of the sample in a temperature range between 20 and 400 °C at a rate of 10 °C/min in an inert atmosphere. Nitrogen was used as a carrier gas at a flow rate of 35 cm^3^/min. The loss in the sample mass was caused by the evaporation of the present water and the degradation of the components in the beads, resulting in the formation of volatile compounds. With the help of thermogravimetric analysis, their content in the beads was determined. Differential scanning calorimetry (DSC) was applied for the determination of the phase transitions of the pure components (Alg, Car, Pol and Cur) and mixtures thereof in order to identify possible interactions of the components. Weighed samples (1.5–3.5 mg) were placed in a hermetically sealed T-aluminum pan and then placed in a DSC instrument (DSC Q-2000, Crawley, UK). The samples were analyzed under a nitrogen atmosphere over a temperature range from 20 °C to a maximum of 250 °C with a heating rate of 10 °C/min. The maximum temperature of the experiment was defined by the TGA study so that decomposition of the material during the DSC analysis was avoided.

### 3.7. Degree of Swelling

The degree of swelling of the carrageenan/alginate as well as carrageenan/alginate/poloxamer beads was determined using the following procedure. A defined mass of dry beads (*m*_0_) was added to the appropriate volume of hydrochloric acid (simulated gastric fluid, pH 1.2) and the phosphate buffer (simulated intestinal fluid, pH 7.4) according to the authors’ study [41]. The temperature was maintained at 37 °C with constant stirring. During the defined time intervals, the swollen beads were separated by filtration, the excess of water was removed with filter paper, and their mass (*m_e_*) was measured. The degree of swelling was determined using Equation (1):(1)$$Swelling\;degree=\frac{m_{e}-m_{0}}{m_{0}}$$

All measurements were performed in triplicate.

### 3.8. Determination of Encapsulation Efficiency

The encapsulation efficiency of curcumin was determined by adding 25.0-mg beads (Car/Alg-Cur or Car/Alg/Pol-Cur) to an appropriate volume of the PBS buffer at a pH value of 7.4 (containing 5 mL ethanol). The final volume was 50 mL [15]. After 24 h, the concentration of encapsulated curcumin was determined by UV/Vis spectrophotometry (Perkin Elmer UV/Vis Lambda 365, Waltham, MA, USA) at a wavelength of 430 nm. The ratio of the spectrophotometrically determined mass of curcumin which was released from the beads and the mass of curcumin initially introduced into the beads represents the encapsulation efficiency (Equation (2)). The measurements were performed in triplicate:(2)$$S\mathrm{EE}\;\left(\%\right)=\frac{\mathrm{Spectrophotometrically}\;\mathrm{determined}\;\mathrm{amount}\;\mathrm{of}\;\mathrm{curcumin}}{\mathrm{Total}\;\mathrm{amount}\;\mathrm{of}\;\mathrm{curcumin}}\times 100$$

### 3.9. In Vitro Release of Curcumin

In the simulation of curcumin release administered via oral route, a change in the concentration of curcumin in hydrochloric acid at pH 1.2 and in the PBS buffer at pH 7.4 was monitored, thereby simulating gastrointestinal tract conditions, according to the authors’ studies [6,41]. The release of curcumin from the beads was followed by using an optimal carrageenan/alginate ratio and optimal concentration of poloxamer 407. The dried, accurately measured mass of beads (50 mg) was added to 50 mL of hydrochloric acid or the PBS buffer and incubated at 37 °C. Curcumin release was monitored over 24 h, where in certain time intervals an aliquot was taken (1.0 mL) in order to spectrophotometrically determine the concentration of released curcumin. The volume of the taken aliquot was replaced with 1.0 mL of fresh HCl/PBS solution. As a control, pure curcumin was weighed and dispersed in the appropriate solutions. Its release was monitored under the same operating conditions as the tested beads. The measurements were performed in triplicate.

### 3.10. The Kinetics of Release

The kinetics of curcumin release from the prepared beads was analyzed by fitting the curcumin release data to the zero-order (Equation (3)), first-order (Equation (4)), Higuchi (Equation (5)), Hixon–Crowell (Equation (6)), Baker–Lonsdale (Equation (7)), and Korsmeyer–Peppas Equation (8)) kinetic equations:

Zero-order kinetic:
(3)$$M_{t}/M_{\infty }=kt$$

First-order kinetic:
(4)$$ln\;\left(M_{t}/M_{\infty }\right)=kt$$

Higuchi model:
(5)$$M_{t}/M_{\infty }=kt^{1/2}$$

Hixon–Crowell model:
(6)$${\left(1-M_{t}/M_{\infty }\right)}^{1/3}=-kt$$

Baker–Lonsdale model:
(7)$$\frac{3}{2}\left[1-{\left(1-\frac{M_{t}}{M_{\infty }}\right)}^{\frac{2}{3}}\right]-\frac{M_{t}}{M_{\infty }}=kt$$

Korsmeyer–Peppas model:
(8)$$M_{t}/M_{\infty }=kt^{n}$$
where $M_{t}/M_{\infty }$ is the fraction of released curcumin at time *t* ($M_{t}$ and $M_{\infty }$ are the amounts of released curcumin at an arbitrary time *t*, expressed in hours, and at equilibrium, respectively), *k* is a constant and *n* is the release exponent indicative of the mechanism of drug release [28,57].

### 3.11. Inflammation Study

In order to analyze the immune response of the Car/Alg/Pol and Car/Alg/Pol-Cur beads, experiments were performed on a macrophage cell line collected from the peritoneal cavities of mice using 5 mL of cold phosphate-buffered saline as previously described [59]. Isolated cells were pelleted for 10 min at 1500 rpm and resuspended in a culture medium of Dulbecco’s modified Eagle medium containing 10% fetal bovine serum. Cells were seeded at a density of 2 × 10^5^ cells/well in 24 well plates and stimulated with lipopolysaccharide (LPS, 1 μg/mL) for 24 h [60]. To determine the influence of the prepared carriers on the macrophage activation, Car/Alg/Pol or Car/Alg/Pol-Cur beads (10, 50 and 100 μg/mL) were incubated with macrophages for 24 h at 37 °C in three independent incubations. After treatment, the cell-free supernatants were collected for nitrite determination. The quantity of nitrite, as an indicator of nitric oxide (NO) production, was measured using Griess reagent [61]. Briefly, 50 μL of culture medium (treated well) was mixed with 50 μL of Griess reagent and was incubated in the dark at room temperature for 10 min. The absorbance was measured at 540 nm using a microplate reader (Zenith 3100, Anthos Labtec Instruments GmbH, Salzburg, Austria).

### 3.12. In Vivo Absorption Studies and CUR Plasma Concentration Analysis

All the animal research studies were conducted in accordance with the guidelines of the Animal Ethics Committee of the Faculty of Medical Sciences at the University of Kragujevac in Serbia (Ethical Approval Number: 01-11805). Eight-week-old male BALB/c mice were housed in cages with hardwood chip bedding under standard laboratory conditions (22 ± 2 °C with a relative humidity of 51 ± 5% and a 12-h light/dark cycle).

The in vivo absorption of pure curcumin and Car/Alg/Pol-Cur beads after oral administration was studied. The mice were deprived of food but had free access to water 12 h before the experiment. The animals were divided into two groups. The first group was orally administered crystalline curcumin (aqueous suspension, 50 mg/kg body weight), and the second group was administered Car/Alg/Pol-Cur beads (mass equivalent to a curcumin dose of 50 mg/kg body weight). After administration, the mice had free access to chow. Blood samples were collected from the abdominal aorta during the euthanasia procedure at different times after treatment (1–24 h). The samples were put into microtubes containing EDTA solution and centrifuged at 13,000 rpm 20 °C for 10 min. The plasma samples (500 μL) were separated, and 1 mL of acetonitrile was added to the plasma [62]. The samples were then centrifuged at 13,000 rpm 20 °C for 5 min, and the supernatant was used for HPLC analysis.

The curcumin concentrations in all plasma samples were determined using an HPLC system (Shimadzu, Kyoto, Japan, diode array detector SPD-M20A) equipped with a C18 column (Hisep column, Supelco, 150 × 4.6 mm, particle size 5 μm) kept at 30 °C. The injection volume was 20 μL, and the mobile phase was acetonitrile: 5.0% acetic acid (50:50 *v*/*v*) with a flow rate of 1.0 mL/min. Curcumin detection was performed at a wavelength of 420 nm. A calibration curve of curcumin was performed after separately mixing the curcumin standard solutions prepared in acetonitrile with a plasma matrix (without curcumin), and the obtained solutions were analyzed by the described HPLC method.

## 4. Conclusions

In this study, curcumin-loaded beads based on cross-linked polysaccharides, κ-carrageenan and sodium alginate were successfully prepared to enhance curcumin bioavailability. In order to increase the percentage of curcumin encapsulation, poloxamer 407 was also added to the beads. The highest percentage of encapsulation (95.74% from 50 mg) of curcumin was obtained using κ-carrageenan/alginate at a ratio of 2:8 and 2.5% of a poloxamer. The results also show the conversion of crystalline curcumin to an amorphous state.

The results of this study illustrate that curcumin bioavailability was significantly improved by the oral administration of the prepared beads, allowing controlled drug release. Additionally, it was concluded that the prepared beads were pH-responsive and that controlled, cumulative drug release took place in the simulated intestinal fluid conditions (PBS, pH 7.4). The highest value of the correlation coefficient was obtained using the Korsmeyer–Peppas model during curcumin release kinetics investigation. The values *n* in the Korsmeyer–Peppas model indicate that the curcumin release from the beads was mediated by the super case II transport mechanism.

Considering the results obtained in this study, it can be underlined that the beads based on biodegradable κ-carrageenan and sodium alginate as well as non-toxic poloxamer 407 can be used as very suitable curcumin carriers, allowing its prolonged retention and efficient and controlled release. Another conclusion is that curcumin beads can be administered orally, with a small percentage of the drug released into the stomach, where undesirable reactions may occur. Furthermore, the prepared beads did not cause any adverse effects in the inflammatory response from the RAW 264.7 macrophage cell line. Finally, the in vivo results suggest that the developed hydrogel beads could be used for the encapsulation and delivery of curcumin and similar bioactive compounds in the food and pharmaceutical industries.

## Figures and Tables

**Figure 1 molecules-27-04045-f001:**
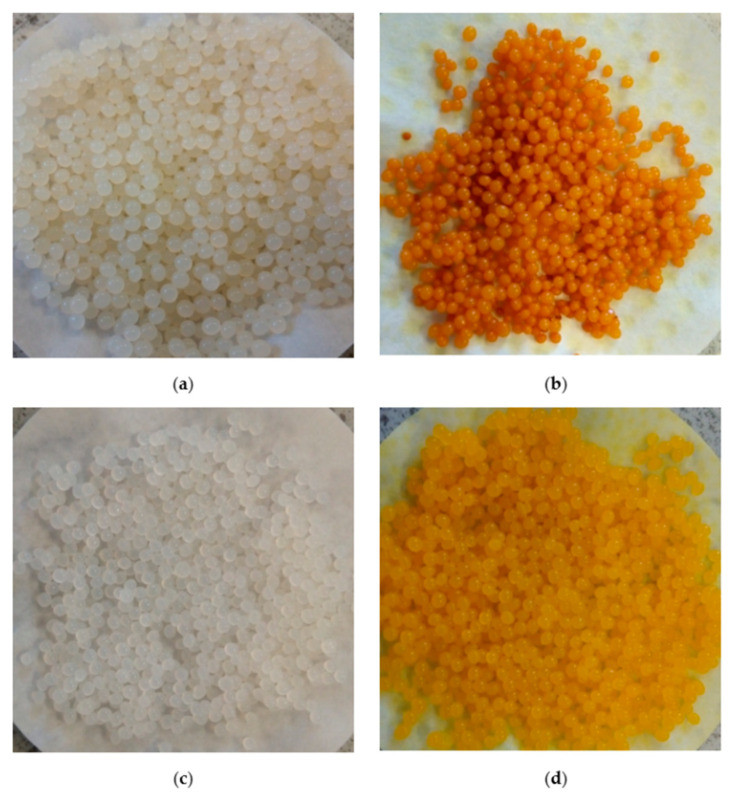
(**a**) Car/Alg beads, (**b**) Car/Alg-Cur beads, (**c**) Car/Alg/Pol beads and (**d**) Car/Alg/Pol-Cur beads.

**Figure 2 molecules-27-04045-f002:**
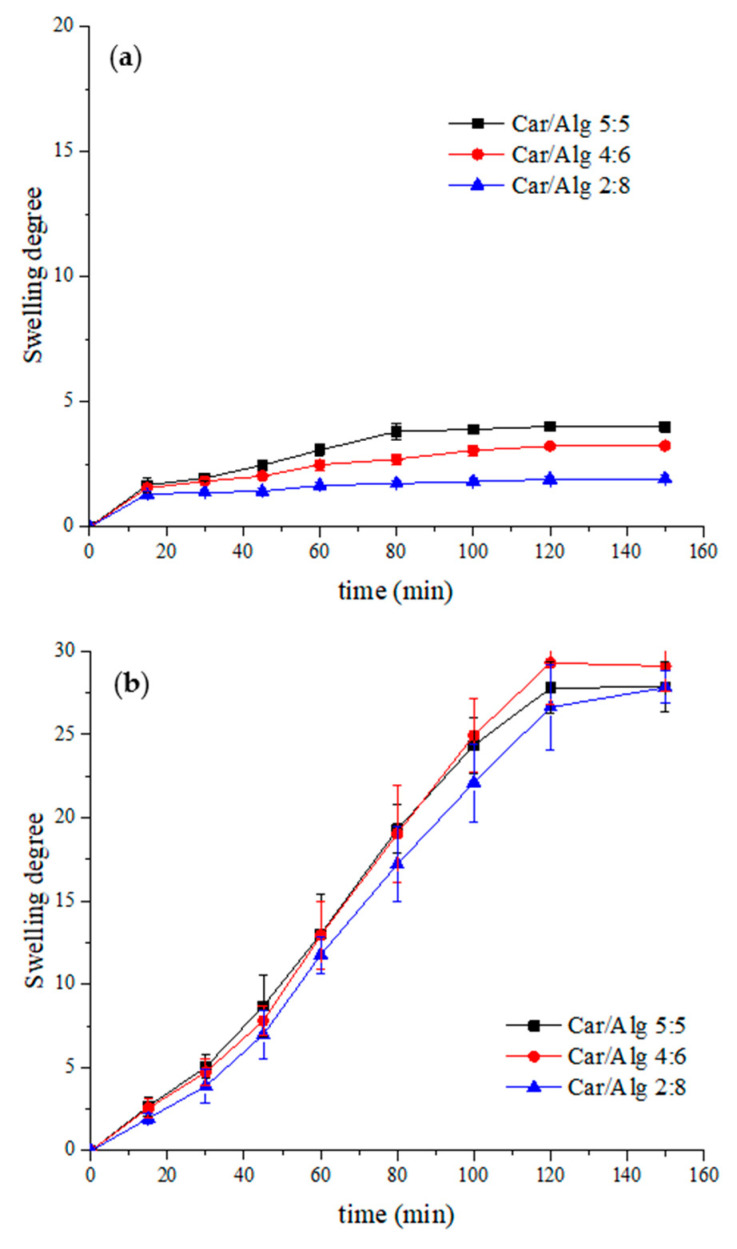
Swelling degree of beads with different carrageenan/alginate ratios in (**a**) HCl solution (pH 1.2) and (**b**) PBS solution (pH 7.4).

**Figure 3 molecules-27-04045-f003:**
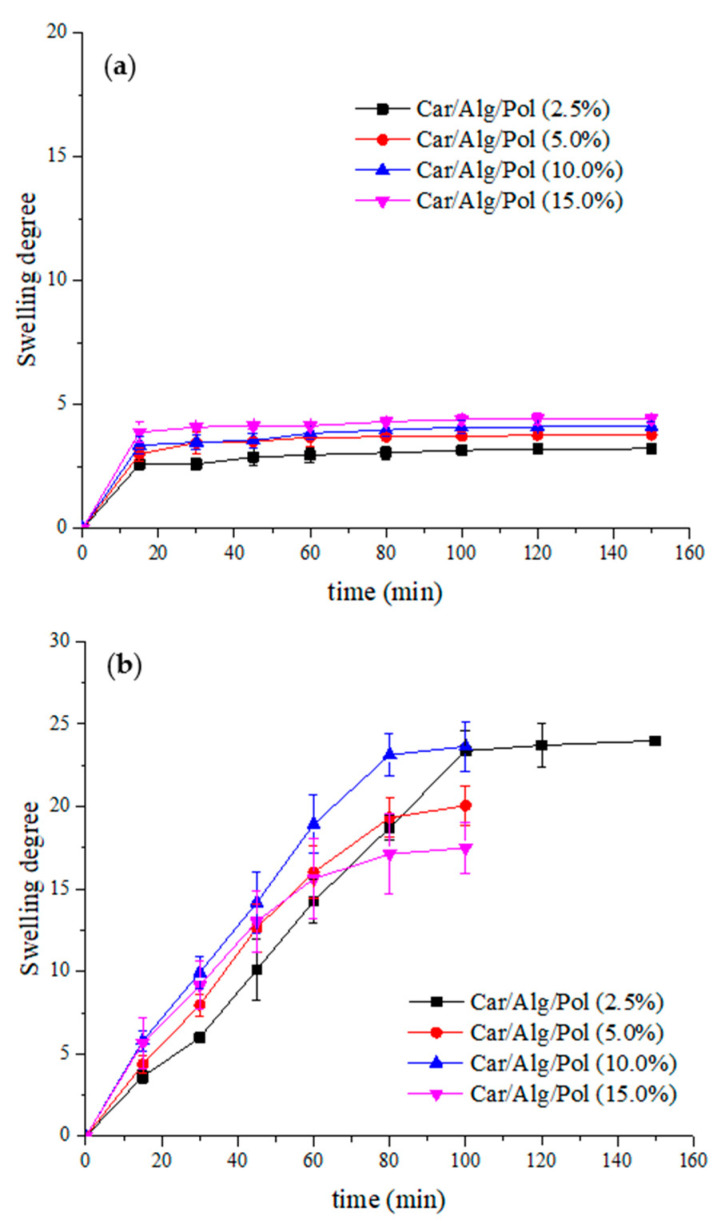
Swelling degree of beads with different poloxamer concentrations in (**a**) HCl solution (pH 1.2) and (**b**) PBS solution (pH 7.4).

**Figure 4 molecules-27-04045-f004:**
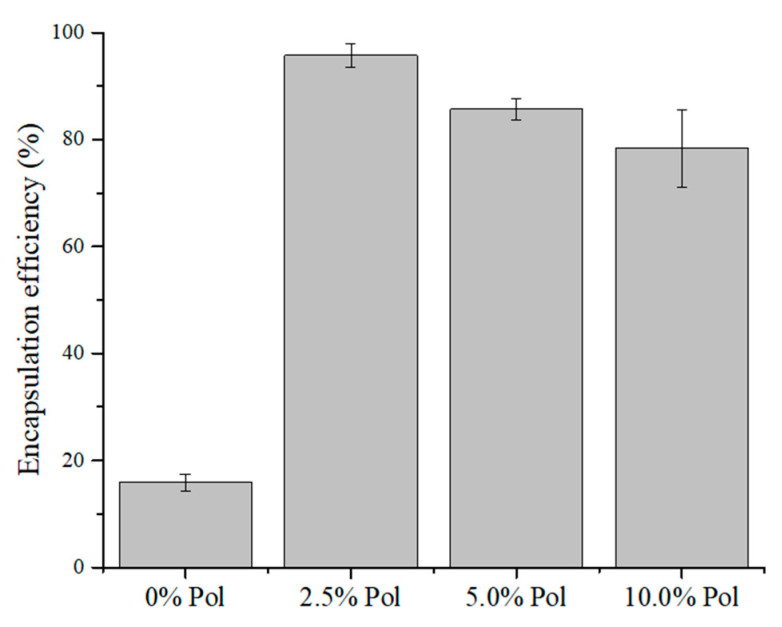
Curcumin encapsulation efficiency.

**Figure 5 molecules-27-04045-f005:**
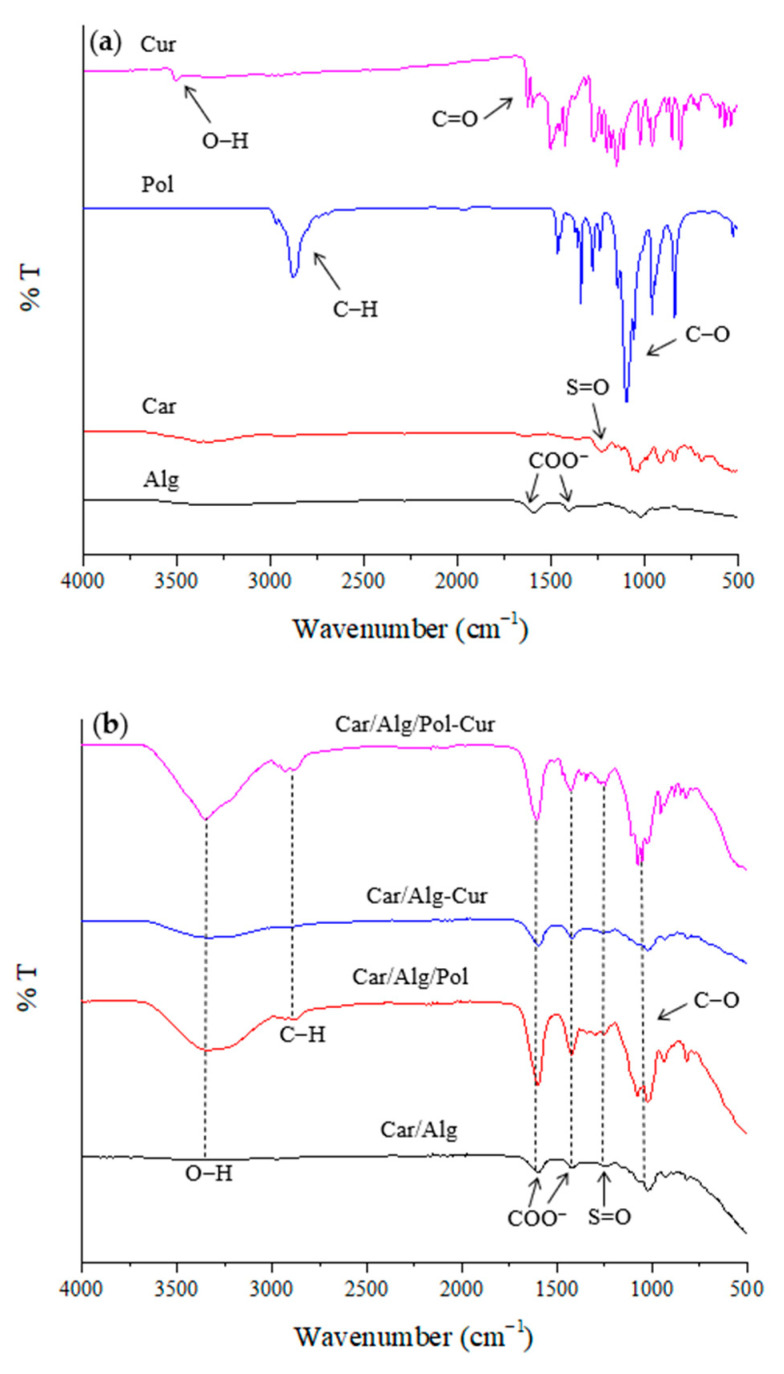
FTIR spectra of (**a**) alginate, κ-carrageenan, poloxamer and curcumin as well as (**b**) Car/Alg, Car/Alg/Pol, Car/Alg-Cur and Car/Alg/Pol-Cur beads.

**Figure 6 molecules-27-04045-f006:**
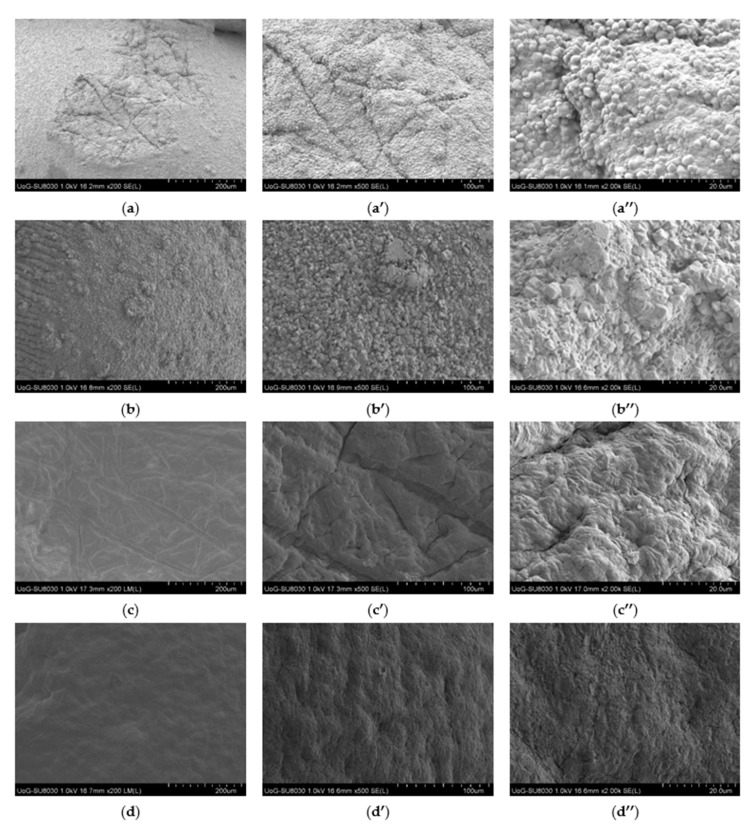
SEM images at 200, 500 and 2000 magnifications of Car/Alg (**a**–**a″**), Car/Alg-Cur (**b**–**b″**), Car/Alg/Pol (**c**–**c″**) and Car/Alg/Pol-Cur beads (**d**–**d″**).

**Figure 7 molecules-27-04045-f007:**
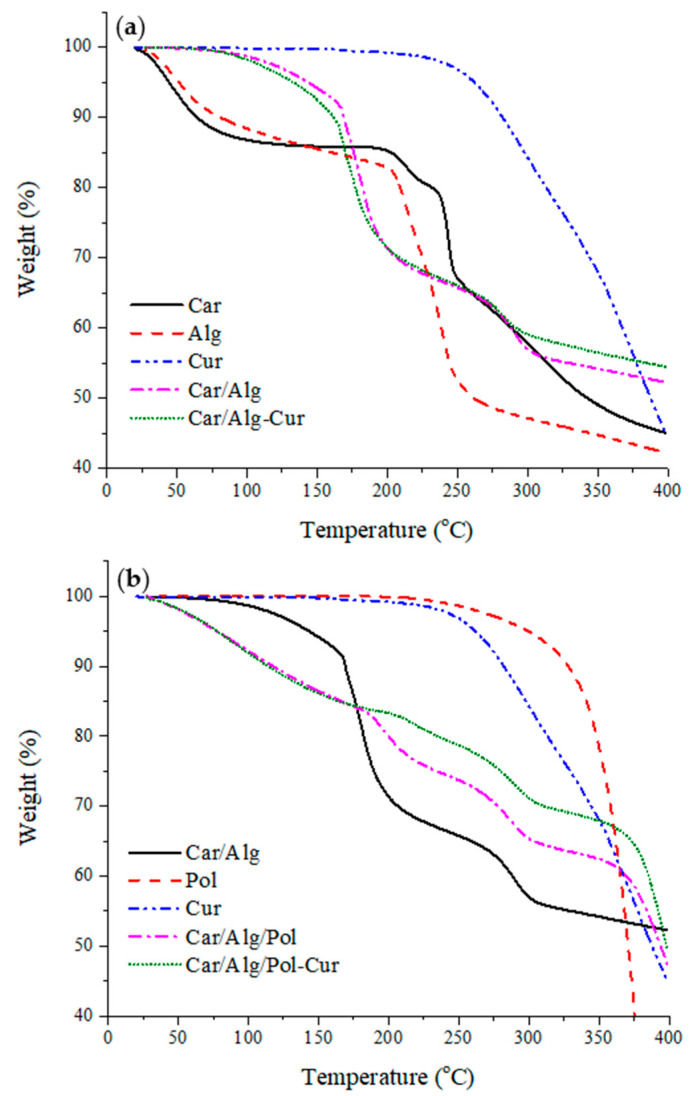
(**a**) TGA thermograms of pure κ-carrageenan, alginate, curcumin, Car/Alg and Car/Alg/-Cur. (**b**) TGA thermograms of pure poloxamer, curcumin, Car/Alg, Car/Alg/Pol and Car/Alg/Pol-Cur beads.

**Figure 8 molecules-27-04045-f008:**
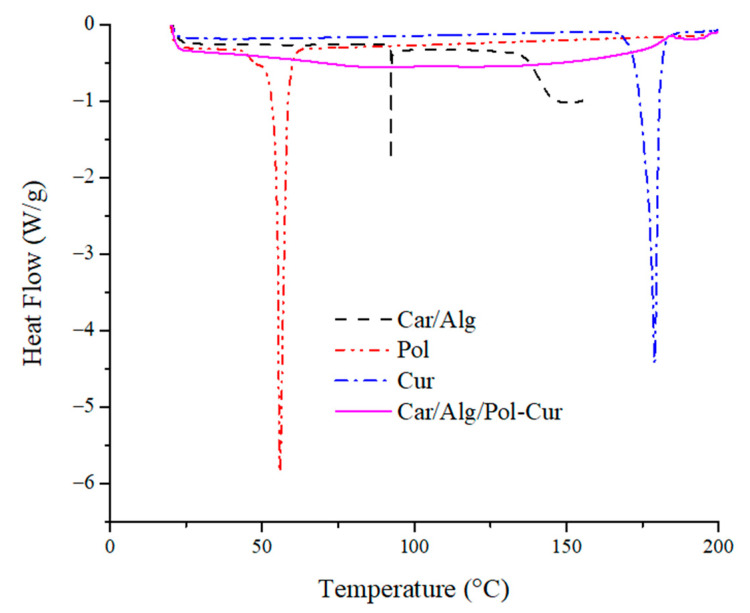
DSC thermograms of pure curcumin, poloxamer, Car/Alg and Car/Alg/Pol-Cur beads.

**Figure 9 molecules-27-04045-f009:**
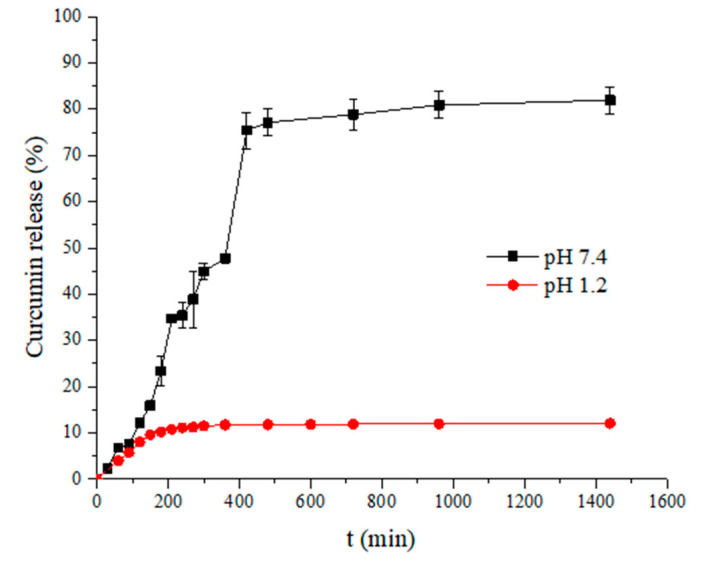
Release profile of curcumin from Car/Alg/Pol-Cur beads in simulated gastrointestinal conditions.

**Figure 10 molecules-27-04045-f010:**
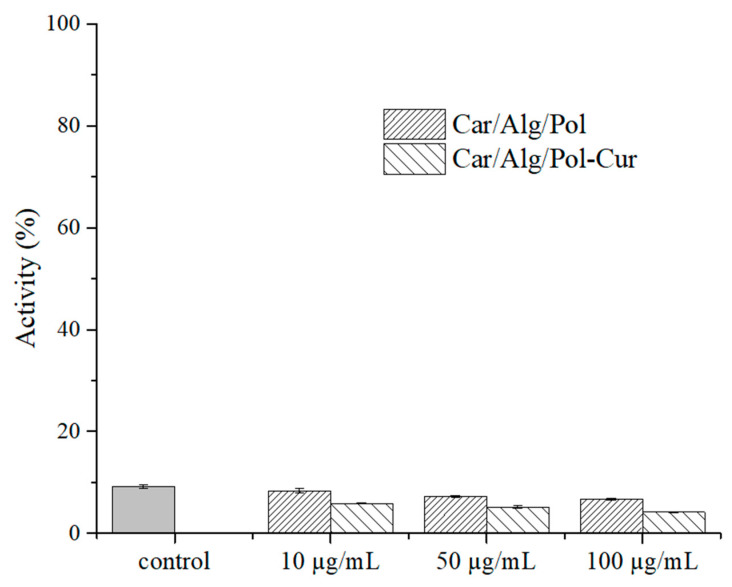
NO produced from activated RAW 264.7 cells after treatment with LPS (control), Car/Alg/Pol and Car/Alg/Pol-Cur hydrogel beads (10–100 μg/mL).

**Figure 11 molecules-27-04045-f011:**
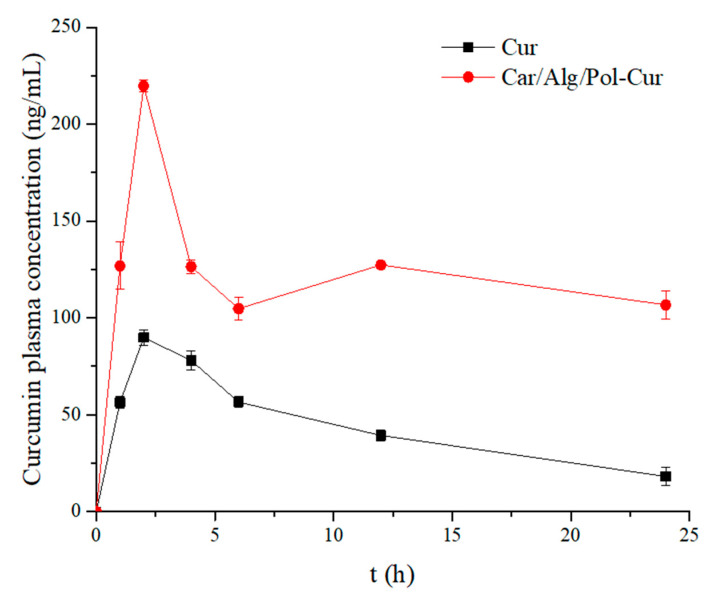
Curcumin plasma concentration in the mice after oral administration of curcumin aqueous suspension and Car/Alg/Pol-Cur beads at a curcumin dose of 50 mg/kg (*n* = 3).

**Table 1 molecules-27-04045-t001:** Characteristics of beads.

Type of Beads	Diameter (mm)	Mass of Curcumin (mg/g of Beads)
Car/Alg	1.10 ± 0.06	/
Car/Alg-Cur	1.03 ± 0.11	10.16 ± 0.15
Car/Alg/Pol	1.23 ± 0.10	/
Car/Alg/Pol-Cur	1.13 ± 0.10	61.21 ± 1.37

**Table 2 molecules-27-04045-t002:** Values of correlation coefficients, rate constants and release exponent.

pH Value	Zero-Order Kinetics		First-Order Kinetics		Highuchi Model		
	*k* _0_	R^2^	*k_I_*	R^2^	*k_H_*		R^2^
1.2	0.0186	0.3362	0.0261	0.2715	0.1325		0.5068
7.4	0.0484	0.7353	0.1205	0.5138	0.2955		0.8815
	**Hixon–Crowell model**		**Baker–Lonsdale model**		**Korsmeyer–Peppas model**		
	*k_HC_*	R^2^	*k_BL_*	R^2^	*k_KP_*	*n*	R^2^
1.2	0.0319	0.7224	0.0186	0.6286	0.4880	1.0023	0.9968
7.4	0.0473	0.9358	0.0265	0.8893	0.1455	1.1451	0.9620

**Table 3 molecules-27-04045-t003:** Pharmacokinetics parameters after oral administration of crystalline curcumin (as an aqueous suspension) and Car/Alg/Pol-Cur beads at a curcumin dose of 50 mg/kg. All values reported are means ± SD (*n* = 3).

Formulation	T_max_ (h)	C_max_ (ng/mL)	AUC (μg·h/mL)
Curcumin	2	89.9 ± 4.1	1.04
Car/Alg/Pol-Cur	2	219.7 ± 2.9	2.91

## Data Availability

Data is contained within the article.

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
