# Peer review of "pH-Responsive Hydrogel Beads Based on Alginate, κ-Carrageenan and Poloxamer for Enhanced Curcumin, Natural Bioactive Compound, Encapsulation and Controlled Release Efficiency"

_molecules, 2022, doi:10.3390/molecules27134045_

Round 1

Reviewer 1 Report

Line 71-72: However, this may also lead to the formation of a precipitate. – explain this sentence; formation of calcium insoluble salts in the presence of certain anions or…?

TGA thermograms- use the same y-scale for both graphs and compare the results.

2.6. The kinetics of release (line 407 and further): Which parameter do you choose to describe the best kinetic model? The table you present show that the Korsmeyer−Peppas model has the highest correlation coefficients. You need to explain your claims (same in the conclusion). Also, can you include the Baker-Lonsdale model (common for spherical matrices)?

Include the reference(s) about poloxamer 407 nontoxicity, since you don’t have a cytotoxicity test in your study.

Reviewer 2 Report

The research work was well planned and executed correctly. In addition to the presented work, the authors need to study the plasma concentration of CUR in mice/rats after oral administration. 

Additionally, the authors need to describe the physical characteristics of the polymers used, namely, molecular weight distribution, M/G ratio for alginate, degree of sulfation for carrageenan etc. 

The inflammatory response of the developed beads needs to be studied as carrageenan is known to cause an inflammatory response.

Round 2

Reviewer 2 Report

The authors have satisfactorily addressed all the comments.